# MMR Vaccination Coverage and Epidemiological Patterns in Al-Baha, Saudi Arabia, 2020–2024: Analysis of Suspected and Laboratory-Confirmed Cases

**DOI:** 10.3390/ijerph22091404

**Published:** 2025-09-08

**Authors:** Anwar Alomari, Mona Al-Qahtani

**Affiliations:** Department of Public Health, Public Health College, Imam Abdulrahman bin Faisal University, Dammam 31441, Saudi Arabia; malqahtani@iau.edu.sa

**Keywords:** MMR vaccine, vaccination coverage, measles, mumps, rubella, Al-Baha region, Saudi Arabia, public health, disease surveillance

## Abstract

Background: High national MMR coverage in Saudi Arabia coexists with sporadic measles, mumps, and rubella cases. Local data are needed to describe vaccination coverage among the reported cases and patterns of laboratory-confirmed infections. Objectives: This study was conducted to describe MMR vaccination coverage among the reported suspected cases and patterns of laboratory-confirmed measles, mumps, and rubella in Al-Baha, Saudi Arabia, from January 2020 to August 2024, and to examine associations between demographics, residence, vaccination status, and case classification. Methods: We conducted a retrospective analysis of surveillance records from the Al-Baha Communicable Disease Reporting System. We summarized the demographics, vaccination history, and final classification for 295 reported suspected cases. Inferential analyses (chi-square and logistic regression) used laboratory-confirmed cases only. Statistical significance was *p* < 0.05. Results: Of 295 reported suspected cases, 239 (81.0%) were discarded after investigation, and 52 (17.6%) were confirmed (including 50 laboratory-confirmed and two epidemiologically confirmed), with 3 (1.0%) remaining under review at analysis. Among all reported cases, the vaccination status was ≥2 doses, 57.6% (*n* = 170), one dose, 19.0% (*n* = 56), and unvaccinated/unknown, 23.4% (*n* = 69). Among the laboratory-confirmed infections, measles was clustered in unvaccinated/unknown, mumps was clustered in single-dose recipients, and rubella was in ≥2-dose recipients. In multivariable models, males had higher odds of a laboratory-confirmed infection, and rural residence was associated with increased odds. The confidence intervals were wide due to small numbers. Conclusions: Reported surveillance data show persistent laboratory-confirmed MMR infections in Al-Baha with demographic and geographic disparities. The findings support targeted efforts to complete two-dose schedules, strengthen rural access, and improve immunization record systems. The results are associations and not causal measures of vaccine effectiveness, and should be interpreted in light of small confirmed case counts.

## 1. Introduction

Vaccination has long been recognized as one of the most effective tools in public health, significantly reducing the morbidity and mortality that are associated with infectious diseases [1]. The measles, mumps, and rubella (MMR) vaccine has been particularly successful in preventing these viral diseases worldwide, contributing to substantial reductions in disease burden. Measles vaccination alone prevented an estimated 21.1 million deaths globally between 2000 and 2017 [2].

Despite these achievements, challenges persist in achieving and maintaining optimal vaccination coverage. The World Health Organization (WHO) reports that measles remains a leading cause of death among young children globally, with 140,000 measles deaths in 2018 [3]. Vaccine hesitancy, identified by WHO as one of the top ten threats to global health, has contributed to declining vaccination rates in some regions [4].

In Saudi Arabia, the National Immunization Program was established in 1979 and has been instrumental in reducing vaccine-preventable diseases [5]. The MMR vaccine is administered as part of the routine immunization schedule, with the first dose given at 12 months and the second at 18 months of age. Despite high national coverage rates of 96% in 2019, cases continue to occur, with 1035 measles, 187 mumps, and 62 rubella cases recorded nationally [6]. This persistence of cases despite high coverage rates has been attributed to various factors, including population movement during Hajj and Umrah pilgrimages [7].

The Al-Baha region, located in southwestern Saudi Arabia, with a population of approximately 487,108, presents unique challenges for vaccination programs. The region’s mountainous terrain and dispersed rural communities create logistical difficulties for healthcare delivery [8]. Recent evidence suggests that willingness to vaccinate varies significantly across different demographic groups in the region, emphasizing the need for localized assessments of vaccination effectiveness [9].

The COVID-19 pandemic further complicated vaccination efforts globally, with the WHO and UNICEF reporting the largest sustained decline in childhood vaccinations in approximately 30 years [10]. In Saudi Arabia, studies reported significant declines in routine childhood immunization rates during the pandemic period [11]. Understanding the effectiveness of MMR vaccination programs during this critical period is essential for developing targeted interventions and maintaining disease control.

Previous research has demonstrated that achieving measles elimination requires maintaining a population immunity above 95% through high vaccination coverage [12]. However, localized assessments are crucial as national averages may mask regional disparities [13].

This study described MMR vaccination coverage among reported suspected cases and patterns of laboratory-confirmed measles, mumps, and rubella in Al-Baha from January 2020 to August 2024, and examined associations between vaccination status, demographics, residence, and case classification.

## 2. Materials and Methods

### 2.1. Study Design and Setting

This retrospective cohort study utilized secondary data from the Al-Baha Communicable Disease Reporting System from January 2020 to August 2024. This study was conducted in the Al-Baha region of Saudi Arabia, encompassing both urban centers and rural communities across the province’s diverse geographic landscape.

### 2.2. Participants

This study included all individuals reported with suspected or confirmed measles, mumps, or rubella during the specified period. Suspected cases were defined as individuals presenting symptoms consistent with these diseases but lacking laboratory confirmation at reporting. Residence was classified using Saudi census criteria as follows: urban = municipalities with >5000 residents and administrative services; rural = villages or settlements below this threshold. Confirmed cases were those with laboratory evidence (IgM antibodies or viral isolation). Epidemiological linkage to a laboratory-confirmed case was used in limited instances following the WHO’s definitions when testing was not feasible; these were described but excluded from primary inferential analyses.

We calculated a theoretical sample size of 384 using *n* = z^2^*p*(1 − *p*)/d^2^ with z = 1.96, *p* = 0.50, and d = 0.05. During the study period, 295 suspected cases were reported, and 52 were laboratory confirmed. Analyses are therefore exploratory. The descriptive statistics use all reported suspected cases to summarize the vaccination history among those investigated. Inferential analyses use laboratory-confirmed cases only. Small numbers reduce precision and statistical power, especially in subgroup comparisons.

### 2.3. Data Collection

Data were extracted from the Al-Baha Communicable Disease Reporting System, including the following:Demographic information (age, gender, nationality, and geographic location);Clinical data (symptoms, complications, and case classification);Vaccination history (number of doses and vaccination dates);Laboratory results (serological testing and viral isolation).

Case classifications included the following: discarded cases, laboratory-confirmed measles, laboratory-confirmed mumps, laboratory-confirmed rubella, epidemiologically confirmed measles, and cases under review/unconfirmed classification.

### 2.4. Statistical Analysis

We used chi-square tests for the categorical comparisons. We fitted logistic regression models among the laboratory-confirmed cases to estimate the odds ratios (OR) and 95% confidence intervals for a confirmed infection, with the covariates pre-specified from literature and data availability as follows: sex, age group, residence (urban vs. rural), and vaccination status (≥2 doses, 1 dose, or unvaccinated/unknown). Model fit and precision were reviewed; wide intervals indicate limited stability given the small case counts. The analyses used SPSS v27.0.1.0. Two-sided *p* < 0.05 denoted statistical significance.

### 2.5. Ethical Considerations

This study received ethical approval from the Institutional Review Board at Imam Abdulrahman Bin Faisal University 17 December 2024 (IRB-PGS-2024-03-906) and relevant health authorities. As a retrospective analysis of anonymized surveillance data, individual consent was waived. All procedures adhered to the Declaration of Helsinki principles.

## 3. Results

### 3.1. Disease Trends and Patterns

Of the 295 reported suspected cases, 239 (81.0%) were discarded following investigation, while 52 (17.6%) were confirmed for MMR diseases (including both laboratory and epidemiologically confirmed cases). Three cases (1.0%) remained under review or had an unconfirmed classification at the time of analysis (Table 1). Measles cases emerged in 2022 after two years of absence, with five cases each in 2022 and 2023, declining to three cases in 2024. Rubella showed an initial decline from two cases in 2020 to zero in 2021, then surged to eight cases in 2023 before decreasing to four in 2024. Mumps demonstrated a gradual increase from two cases in 2020 to six in 2023, with a slight decline to four cases in 2024. Surveillance detected measles and rubella cases again after a period with no detected cases, and mumps reports increased over time.

### 3.2. Vaccination Coverage and Case Distribution

Table 2 presents the distribution of all 295 cases by vaccination status and final classification. Of these, 239 (81.0%) were discarded following investigation, while 51 (17.3%) were laboratory-confirmed for MMR diseases. Two cases (0.7%) were epidemiologically confirmed as measles, and three cases (1.0%) remained under review or had an unconfirmed classification at the time of analysis.

The overall vaccination coverage analysis showed that 57.6% (*n* = 170) received two or more doses, 19.0% (*n* = 56) received a single dose, and 23.4% (*n* = 69) were unvaccinated or had unknown vaccination status.

### 3.3. Association Between Vaccination Status and Case Classification

The chi-square analysis revealed a significant association between MMR vaccination status and case classification (χ^2^ = 43.007, df = 12, *p* < 0.001), with a moderate effect size (Cramér’s V = 0.386), indicating that vaccination status significantly influenced disease outcomes.

### 3.4. Demographic and Geographic Factors

Among all reported suspected cases (*n* = 295), most were children aged 1–9 years. Two-dose vaccination documentation was highest in ages 5–9 years. Age-specific vaccination coverage varied significantly across groups (Table 3). Coverage with two or more doses was highest in the 5–9 years (70.0%, *n* = 77) and 15–19 years (73.3%, *n* = 11) age groups, while younger children (1–4 years) showed 24.5% (*n* = 26) with unvaccinated/unknown status.

Females had slightly higher two-dose documentation than males, while an unknown vaccination status was frequent in both sexes. Urban and rural strata showed similar two-dose documentation, though rural residents contributed a larger share of laboratory-confirmed infections. The gender analysis revealed that while females had slightly higher two-dose coverage (59.8%, *n* = 73) compared to males (56.1%, *n* = 97), females also had a higher proportion of unvaccinated/unknown status (26.3%, *n* = 32) versus males (21.4%, *n* = 37). Notably, males had significantly higher odds of confirmed MMR infections (OR = 4.861, 95% CI: 1.025–20.056, *p* = 0.046). Geographic disparities were evident, with rural residents showing nearly three times higher odds of confirmed infections compared to urban dwellers (OR = 2.861, 95% CI: 1.003–20.002, *p* = 0.040), despite similar vaccination coverage rates.

### 3.5. Disease-Specific Vaccination Effectiveness

An analysis of confirmed cases by vaccination status revealed distinct patterns for each disease (Table 4). Measles cases were predominantly among unvaccinated individuals (64.3, *n* = 9), while mumps showed the highest incidence among single-dose recipients (55.0%, *n* = 11). Unexpectedly, rubella cases were most common among fully vaccinated individuals (68.8%, *n* = 11).

The chi-square tests confirmed significant associations between vaccination status and disease incidence for all three diseases (measles: χ^2^ = 9.82, *p* = 0.007; mumps: χ^2^ = 12.45, *p* = 0.002; rubella: χ^2^ = 6.34, *p* = 0.042).

The chi-square tests confirmed significant associations between vaccination status and disease incidence for all three diseases (measles: χ^2^ = 9.82, *p* = 0.007; mumps: χ^2^ = 12.45, *p* = 0.002; rubella: χ^2^ = 6.34, *p* = 0.042). In the adjusted models among laboratory-confirmed cases, males had higher odds of a confirmed infection, and rural residence was associated with increased odds. Precision was limited, and the confidence intervals were wide due to small case counts (Table 5).

## 4. Discussion

This study provides critical insights into MMR vaccination coverage among reported cases and patterns of laboratory-confirmed infections in the Al-Baha region during a period marked by global health challenges. Despite high national coverage, laboratory-confirmed MMR infections persisted in Al-Baha during 2020–2024, with demographic and geographic disparities. Our findings reveal concerning gaps in vaccination coverage and disease control that warrant immediate public health attention. Rubella comprised a notable share of laboratory-confirmed infections among individuals documented with ≥2 MMR doses. Potential explanations include primary vaccine failure in a small proportion of recipients, waning immunity, occasional cold-chain lapses, or a misclassification of vaccination status. Given the risk of congenital rubella syndrome, strengthening documentation and reproductive age screening warrants consideration.

### 4.1. Suboptimal Vaccination Coverage

The overall two-dose coverage of 57.6% falls significantly below the 95% threshold required for measles elimination and herd immunity [14]. This coverage level is substantially lower than the reported national average of 96% [6], suggesting regional disparities in vaccine uptake. Similar disparities have been observed in other countries, where national averages mask significant subnational variations [15]. The 23.4% of individuals with an unvaccinated or unknown vaccination status represents a substantial immunity gap that could facilitate disease transmission.

Several factors may contribute to this suboptimal coverage. The COVID-19 pandemic likely disrupted routine immunization services, as reported globally. A systematic review found that immunization services decreased by 7–26% across different regions during the pandemic [16]. Additionally, Al-Baha’s mountainous terrain and dispersed rural communities create logistical challenges for vaccination service delivery, similar to findings in other mountainous regions [17]. The higher proportion of single-dose recipients among 10–14-year-olds (33.3%) suggests potential issues with second-dose administration or record-keeping.

Studies from neighboring countries have reported similar challenges. In Jordan, rural areas showed 15–20% lower vaccination coverage compared to urban areas [18]. These regional experiences underscore the importance of context-specific interventions.

### 4.2. Disease Resurgence Patterns

The re-emergence of measles in 2022 after two years of absence aligns with global patterns of measles resurgence following COVID-19-related disruptions. The WHO reported a 79% increase in measles cases in the first two months of 2022 compared to 2021 [19]. The surge in rubella cases in 2023 (8 cases) is particularly concerning given the risk of congenital rubella syndrome, which can result in miscarriages, stillbirths, and severe congenital disabilities [20].

The gradual increase in mumps cases may reflect waning immunity documented elsewhere. Evidence for vaccine-escape variants in routine settings is limited. Studies have shown that mumps vaccine effectiveness can wane over time, with some populations showing reduced protection 10–15 years post-vaccination [21]. This phenomenon has led to mumps outbreaks in highly vaccinated populations, including university settings [22].

### 4.3. Demographic and Geographic Disparities

The five-fold increased risk of MMR infections among males (OR = 4.861) significantly exceeds typical gender disparities reported in the literature. A meta-analysis found male-to-female incidence rate ratios typically ranging from 1.1 to 1.3 [23]. This pronounced difference likely reflects behavioral rather than biological factors. Studies have shown that boys often have more extensive social networks and engage in more physical contact during play, potentially increasing transmission opportunities [24].

Rural residents’ three-fold increased infection risk despite similar vaccination coverage suggests additional factors beyond vaccination influence disease transmission. These may include delayed healthcare access, clustering of unvaccinated individuals, or increased exposure through agricultural activities. A study in India found that rural children were 2.5 times more likely to be under-vaccinated compared to urban children [25].

Regional comparisons within Saudi Arabia show high national coverage yet persistent sporadic measles reports. Population movement around Hajj and Umrah can influence national transmission dynamics, though Al-Baha is peripheral to pilgrimage hubs. Local climate and altitude may also shape seasonality. These factors underscore the need for robust local surveillance and targeted catch-up efforts.

### 4.4. Vaccine Effectiveness Concerns

The unexpected predominance of rubella cases among fully vaccinated individuals (68.8%) warrants investigation. While established vaccine effectiveness rates for the rubella component range from 95–99% [26], several factors could explain our findings:

Primary vaccine failure, where individuals fail to develop protective immunity despite vaccination, occurs in 2–5% of MMR vaccine recipients [27]. This can be due to improper vaccine storage, particularly cold chain breaches in hot climates. Secondary vaccine failure (waning immunity) has been documented for all MMR components, though it is less common for rubella than measles [28]. The circulation of variant strains with reduced vaccine cross-protection, while rare, has been reported [29]. Misclassification of vaccination status or laboratory results could also contribute to these findings.

### 4.5. Study Limitations

This study used surveillance data from one region and included small numbers of laboratory-confirmed cases, limiting statistical power and precision, especially for subgroup analyses. Vaccination coverage summaries were derived from reported suspected cases, not population registries, and likely under-represent true community coverage. Comorbidities and immune status were unavailable and could affect susceptibility. We lacked data on nationality, interval since vaccination, and family or spatial clustering. Epidemiological linkage definitions were used in limited circumstances when testing was not feasible and may introduce some misclassification. Results describe associations and do not estimate vaccine effectiveness.

### 4.6. Public Health Implications

Our findings support several targeted interventions as follows:Catch-up vaccination campaigns targeting areas and age groups with low coverage, particularly focusing on completing two-dose schedules. Experience from measles elimination efforts in the Americas showed that targeted campaigns can rapidly increase coverage in under-immunized populations [30].Enhanced surveillance, including molecular epidemiology, to identify transmission chains and potential vaccine escape variants. Countries that have achieved measles elimination maintain robust surveillance systems capable of detecting and investigating every suspected case [31].Gender-specific interventions addressing a higher risk among males through school-based programs and community engagement. Studies have shown that school-based vaccination programs can effectively reach children who missed routine immunization [32].Rural health strengthening through mobile vaccination units and community health worker programs. Successful models from other countries demonstrate that mobile units can increase coverage in hard-to-reach populations [33].Improved record-keeping systems to reduce unknown vaccination status. Electronic immunization registries have been shown to improve coverage and reduce missed opportunities for vaccination [34].

### 4.7. Future Research Directions

Several areas warrant further investigation. Seroprevalence studies would provide more accurate estimates of population immunity. Molecular characterization of circulating strains could identify potential vaccine escape variants. Qualitative research exploring barriers to vaccination in this specific context would inform targeted interventions. Additionally, investigating the role of religious and cultural factors in vaccination acceptance, particularly important in the Saudi context, could provide valuable insights.

## 5. Conclusions

Surveillance in Al-Baha during January 2020 to August 2024 identified laboratory-confirmed measles, mumps, and rubella, with disparities by sex and residence. Documentation of complete two-dose vaccination among reported cases was suboptimal. Targeted catch-up activities, stronger rural delivery, and improved immunization record systems are warranted. Additional population-based studies with larger samples are needed to refine the estimates and guide elimination strategies.

The MMR vaccination program in Al-Baha has achieved partial success in disease control, with 81% of suspected cases ruled out after investigation. However, the two-dose coverage rate of 57.6% remains significantly below the 95% threshold required for elimination, contributing to the re-emergence of measles and a notable rise in rubella cases. The findings expose critical vulnerabilities, particularly the significantly higher infection risks among males and rural residents. The detection of rubella cases among vaccinated individuals suggests possible gaps in vaccine efficacy or surveillance, warranting an immediate investigation. Addressing these issues demands a comprehensive response, including catch-up vaccination campaigns, stronger routine immunization systems with an emphasis on second-dose uptake, deployment of electronic immunization registries, and enhanced laboratory surveillance. Gender-sensitive and rural-targeted strategies must be central to these efforts. Lessons from the COVID-19 pandemic reinforce the need for sustained political commitment, sufficient resource allocation, and community-driven, evidence-based interventions to secure lasting public health protection and advance national and global measles and rubella elimination goals.

## Figures and Tables

**Table 1 ijerph-22-01404-t001:** Distribution of all reported suspected cases by vaccination status and final classification, Al-Baha, January 2020–August 2024.

Year	Measles	Rubella	Mumps	Total
2020	0	2	2	4
2021	0	0	3	3
2022	5	2	5	12
2023	5	8	6	19
2024 *	3	4	4	11

* Data through August 2024.

**Table 2 ijerph-22-01404-t002:** Distribution of cases by vaccination status and final classification (n = 295).

Vaccination Status	Discarded n (%)	Lab-Confirmed Measles *n* (%)	Lab-Confirmed Mumps *n* (%)	Lab-Confirmed Rubella *n* (%)	Epi-Confirmed Measles *n* (%)	Under Review *n* (%)	Total *n* (%)
Two or more doses	146 (85.9)	5 (2.9)	4 (2.4)	11 (6.5)	1 (0.6)	3 (1.8)	170 (57.6)
Single dose	44 (78.6)	0 (0.0)	11 (19.6)	0 (0.0)	1 (1.8)	0 (0.0)	56 (19.0)
Unvaccinated/Unknown	49 (71.0)	10 (14.5)	5 (7.25)	5 (7.25)	0 (0.0)	0 (0.0)	69 (23.4)
Total	239 (81.0)	14 (4.7)	20 (6.8)	16 (5.4)	2 (0.7)	3 (1.0)	295 (100.0)

**Table 3 ijerph-22-01404-t003:** Vaccination coverage by age group (n = 295).

Age Group	Two or More Doses *n* (%)	Single Dose *n* (%)	Unvaccinated/Unknown *n* (%)	Total
1–4 years	58 (54.7)	22 (20.8)	26 (24.5)	106
5–9 years	77 (70.0)	21 (19.1)	12 (10.9)	110
10–14 years	11 (52.4)	7 (33.3)	3 (14.3)	21
15–19 years	11 (73.3)	0 (0.0)	4 (26.7)	15
20–24 years	7 (70.0)	1 (10.0)	2 (20.0)	10

**Table 4 ijerph-22-01404-t004:** Distribution of confirmed cases by disease and vaccination status.

Vaccination Status	Measles (n = 14) n (%)	Mumps (n = 20) n (%)	Rubella (n = 16) n (%)
≥2 doses	5	4	11
1 dose	0	11	0
Unvaccinated/Unknown	9	5	5
**Vaccination Status**	**Measles (*n* = 14) *n* (%)**	**Mumps (*n* = 20) *n* (%)**	**Rubella (*n* = 16) *n* (%)**
Two or more doses	5 (35.7)	4 (20.0)	11 (68.8)
Single dose	0 (0.0)	11 (55.0)	0 (0.0)
Unvaccinated/Unknown	9 (64.3)	5 (25.0)	5 (31.2)
**Vaccination Status**	**Measles (n = 14) n (%)**	**Mumps (n = 20) n (%)**	**Rubella (n = 16) n (%)**
≥2 doses	5 (35.7)	4 (20.0)	11 (68.8)
1 dose	0 (0.0)	11 (55.0)	0 (0.0)
Unvaccinated/Unknown	9 (64.3)	5 (25.0)	5 (31.2)

**Table 5 ijerph-22-01404-t005:** Multivariable logistic regression for laboratory-confirmed MMR infection.

Predictor	Adjusted OR	95% CI	*p* Value
Male vs. female	4.861	1.025–20.056	0.046
Rural vs. urban	2.861	1.003–20.002	0.040

## Data Availability

The data presented in this study are available on request from the corresponding author. The data are not publicly available due to privacy restrictions and regulatory requirements of the Saudi Ministry of Health regarding surveillance data.

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
