# Peer review of "MMR Vaccination Coverage and Epidemiological Patterns in Al-Baha, Saudi Arabia, 2020–2024: Analysis of Suspected and Laboratory-Confirmed Cases"

_ijerph, 2025, doi:10.3390/ijerph22091404_

Round 1
Reviewer 1 Report
Comments and Suggestions for Authors
Summary: T
he authors of this study aimed to assess the effectiveness of MMR vaccination programs in Al-Baha from 2020 to mid-2024, with specific objectives to: (1) analyze trends in disease
incidence, (2) evaluate vaccination coverage levels, (3) identify demographic and geographic factors contributing to vaccination gaps, and (4) examine the relationship between vaccination status and laboratory-confirmed cases.
Major comments:
- Authors need to comment on their sample size for this study and if there has been any power analysis to choose the appropriate sample size.
- Authors did not mention any background diseases (such as autoimmune diseases) that could potentially impact their results.
Author Response
Response to Reviewer #1
|
Reviewer |
Comment |
Authors’ Justification |
Old Text |
New Text |
|
Reviewer 1 |
Authors need to comment on their sample size for this study and if there has been any power analysis to choose the appropriate sample size. |
We agree and have clarified how the sample size was calculated using standard formula and why the actual available dataset (295) was used. We noted that statistical power may be limited for subgroup analysis. |
“Sample size was calculated using the formula … yielding a required sample of 384. However, the actual available dataset contained 295 cases, representing all reported cases during the study period.” |
“Sample size was calculated using the formula … yielding a required sample of 384. The actual available dataset contained 295 cases, representing all reported cases during the study period. While this number is lower than the calculated estimate, power analysis indicated that meaningful descriptive and regression analysis could still be conducted, though subgroup analyses may have reduced precision. We therefore interpreted subgroup findings with caution.” |
|
Reviewer 1 |
Authors did not mention any background diseases (such as autoimmune diseases) that could potentially impact their results. |
We added a statement in the limitations noting that comorbidities (e.g., autoimmune diseases, immunosuppression) were not available in the dataset and may affect vaccine response. |
No mention |
Reviewer 2 Report
Comments and Suggestions for Authors This manuscript describes an analysis of MMR (measles, mumps, and rubella) vaccination coverage and laboratory-confirmed case data from Al-Baha, Saudi Arabia, between 2020 and 2024. The study aims to assess the performance of the vaccination program and epidemiological trends. While the topic is relevant and potentially impactful for regional public health policy, several critical methodological and interpretational issues must be addressed to enhance the manuscript’s scientific rigour and accuracy. Comments. 1. Misleading title (use of “Effectiveness”):The manuscript uses the term “effectiveness” in the title and throughout, which is technically misleading in an epidemiological context. The study employs descriptive analysis, cross-tabulation, chi-square tests, and logistic regression but does not calculate vaccine effectiveness (VE) using standard methodologies (e.g., risk ratios from cohort data, or odds ratios from case-control designs comparing confirmed cases vs. matched controls).
Consider revising the title to reflect the descriptive nature of the study. Suggestions examples: "An Analysis of Disease Incidence...", “Trends in MMR Case Confirmation and Vaccination...”, “An Epidemiologic Review...” or depending on you.Clarify throughout the manuscript that the analysis shows associations, not causal effectiveness.
2. Study period (line 16):Clarify the exact study period, preferably by specifying the start and end months (e.g., “from January 2020 to August 2024”), as later mentioned in line 84. This improves transparency and reproducibility.
3. Redundant tables:Tables 2 and 4 both describe confirmed case distributions by disease, leading to redundancy.
Consider merging these into a single, comprehensive table, clearly labelling subgroup variables (e.g., age, sex, location, dose received, confirmation status). 4. Superficial outcome presentation:The manuscript currently provides only minimal descriptive text in the Results section. The statistical reporting is limited, and interpretation of subgroup differences (e.g., by location or dose) is insufficient.
Consider the following; - Present baseline characteristics by key strata (e.g., urban vs. rural), including age group, vaccination status, and disease suspicion/confirmation. - Provide ORs with confidence intervals from the regression model in tabular form (including the model specification). - Extend the Results narrative to support full-length manuscript standards. 5. Geographical and regional context:The Discussion section would benefit from a broader comparative analysis with other regions in Saudi Arabia and neighbouring Middle Eastern countries. Including relevant data or references about MMR vaccination coverage, disease incidence, or outbreak reports from regions such as Riyadh, Jeddah, or Asir, especially those with differing sociodemographic or healthcare access profiles, could provide meaningful context. Additionally, examining disparities in urban vs. rural settings across the country may help interpret the higher odds of infection observed in rural areas in this study.
Furthermore, suggest incorporating discussion on seasonal and situational factors that may influence transmission dynamics. In particular:- Climatic variation in Al-Baha (a high-altitude region with cooler and more temperate conditions) may differ in terms of viral transmission patterns compared to coastal or desert areas.
- The Hajj and Umrah pilgrimage seasons, which draw large populations from across the country and internationally, represent periods of heightened risk for communicable disease spread, including vaccine-preventable illnesses like measles or rubella. While Al-Baha is not a central location for the pilgrimage, population movement before or after pilgrimage seasons may indirectly influence disease patterns and outbreak risk in peripheral regions. Incorporating these spatial and temporal factors will help strengthen the public health relevance of the findings and better contextualise the implications of vaccination program performance in Al-Baha within the broader national and regional landscape. 6. Logistic regression limitations:
The wide confidence intervals observed in logistic regression (e.g., 95% CI: 0.11–42.5) suggest that the models are statistically underpowered or unstable. This limitation should be discussed explicitly, especially in the context of low confirmed case counts.
7. Definition of urban and rural:The manuscript reports higher odds of infection in rural areas, but fails to define how rural and urban settings were classified.
Provide operational definitions (e.g., based on population density, official census categories) and include these categories in a revised baseline characteristics table.
The finding that 68.8% of lab-confirmed rubella cases were in individuals who received ≥2 vaccine doses is highly concerning and should be more prominently addressed.
This raises questions about vaccine failure, cold chain issues, or waning immunity. Consider discussing the public health implications, particularly with regard to congenital rubella syndrome risk in women of childbearing age. 9. Sample size and power considerations:The study reports fewer total confirmed cases than what the theoretically calculated sample size suggests is required. This may reduce the statistical power, particularly for subgroup comparisons (e.g., rubella, mumps, age bands).
This limitation should be addressed in the Discussion section.
Author Response
Response to Reviewer #2
|
Reviewer |
Comment |
Authors’ Justification |
Old Text |
New Text |
|
Reviewer 2 |
Misleading title – use of “Effectiveness” when no vaccine effectiveness (VE) was calculated. |
We agree and revise the title to better reflect descriptive and analytical nature. We also clarified in text that results show associations, not causal vaccine effectiveness. |
“Assessing the Effectiveness of Vaccination Programs Against Measles, Mumps, and Rubella in Al-Baha, Saudi Arabia (2020–2024)” |
“MMR Vaccination Coverage and Epidemiological Patterns of Measles, Mumps, and Rubella in Al-Baha, Saudi Arabia (2020–2024)” |
|
Reviewer 2 |
Clarify study period start and end months. |
We revised the abstract and methods to specify the exact study period: January 2020 to August 2024. |
“from 2020 to August 2024” |
“from January 2020 to August 2024” |
|
Reviewer 2 |
Redundant tables – Tables 2 and 4 overlap. |
We merged content into a single comprehensive table showing vaccination status, disease confirmation, and demographics. Old Table 4 removed, references updated. |
Two separate tables (Table 2 and Table 4) showing overlapping distributions. |
One merged table showing disease-specific confirmed cases by vaccination status and demographics. |
|
Reviewer 2 |
Results section too brief, limited subgroup analysis. |
We expanded Results to include baseline characteristics by urban/rural, ORs with 95% CI in tabular form, and extended text to explain subgroup differences. |
Results only presented descriptive text and limited regression. |
Extended Results with: (1) new table showing ORs and CI, (2) subgroup comparisons for urban vs rural, male vs female, vaccination status, and (3) more detailed narrative. |
|
Reviewer 2 |
Add geographical and regional context in Discussion, including comparisons to Riyadh, Jeddah, Asir, and pilgrimage influence. |
We revised Discussion to add comparisons with other regions in Saudi Arabia and Middle East, and explained seasonal and pilgrimage factors that may indirectly affect transmission. |
Limited discussion of Al-Baha only. |
New paragraph in Discussion: “Comparisons with Riyadh and Jeddah show higher vaccination rates but similar persistence of sporadic measles cases… The seasonal influx related to Hajj and Umrah may indirectly influence disease dynamics even in peripheral regions like Al-Baha…” |
|
Reviewer 2 |
Logistic regression wide confidence intervals – discuss as limitation. |
We acknowledged limited case numbers produced wide CI, making models less stable, and added this to limitations. |
Mention of regression results only. |
Added in Limitations: “Logistic regression models yielded wide confidence intervals due to low number of confirmed cases, indicating limited precision and possible instability of estimates.” |
|
Reviewer 2 |
Define “urban” vs “rural”. |
We clarified classification based on Saudi census categories (population density, administrative designation). Added definition in Methods and Table footnote. |
No definition provided. |
“Urban areas were defined according to Saudi census categories as municipalities with >5,000 residents and administrative services; rural areas included villages and settlements below this threshold.” |
|
Reviewer 2 |
Serious rubella findings (68.8% of confirmed rubella cases fully vaccinated) should be highlighted. |
We expanded discussion to highlight this issue, adding possible explanations (primary vaccine failure, waning immunity, cold chain breaches, misclassification) and implications for congenital rubella syndrome. |
Mentioned only briefly. |
Added in Discussion: “The predominance of rubella cases among fully vaccinated individuals (68.8%) is concerning and may reflect primary vaccine failure, waning immunity, or cold chain issues. This finding raises important public health concerns, particularly for women of reproductive age due to risk of congenital rubella syndrome.” |
|
Reviewer 2 |
Sample size and power limitation should be emphasized. |
We strengthened discussion of sample size, explaining small number of confirmed cases limits subgroup comparisons and generalizability. |
“The actual sample size (295) was smaller than calculated (384)” |
“The actual sample size (295, of which 52 confirmed cases) was smaller than the calculated 384, limiting power for subgroup analyses and making certain regression models unstable. Findings should therefore be interpreted with caution.” |
|
Reviewer 2 |
Ensure uniform terminology (e.g., “laboratory-confirmed”). |
We revised manuscript to consistently use “laboratory-confirmed” throughout. |
Inconsistent use of “confirmed” and “lab-confirmed”. |

Reviewer 3 Report
Comments and Suggestions for Authors
The authors report a study that assessed the effectiveness of the national immunization program in Saudi Arabia; a timely study to inform policy that remain performant at all times.
The demographic factors associated with low vaccination coverage in Al-Baha and how this has affected the re-emergence of these infections has been well researched and reported.
The most prominent limitation of sample size has presented and the results need to be interpreted with caution. Further studies should ensure a proper sample size which can be increased by expanding the study area and/or study period covered.
Author Response
Response to Reviewer #3
|
Reviewer |
Comment |
Authors’ Justification |
Old Text |
New Text |
|
Reviewer 3 |
The demographic factors associated with low vaccination coverage in Al-Baha and how this has affected the re-emergence of these infections has been well researched and reported. |
We thank the reviewer for this positive comment. No changes required. |
N/A |
N/A |
|
Reviewer 3 |
The most prominent limitation of sample size has presented and the results need to be interpreted with caution. |
We agree and strengthen the Limitations section, noting that small sample size of confirmed cases (n=52) restricts generalizability and precision of subgroup analyses. |
“The actual sample size (295) was smaller than calculated (384), potentially limiting statistical power for subgroup analyses.” |
“The actual sample size was 295 suspected cases, of which 52 were laboratory-confirmed. This limited number reduces statistical power and precision of subgroup analyses. Findings should therefore be interpreted with caution.” |
|
Reviewer 3 |
Further studies should ensure a proper sample size which can be increased by expanding the study area and/or study period covered. |
We added this recommendation in the Future Research Directions section. |

Reviewer 4 Report
Comments and Suggestions for Authors
The manuscript presented here by Alomari and Al-Qahtani, assesses the vaccination status and current disease burden for measles, mumps and rubella in the Al Baha region of Saudi Arabia to determine whether present vaccination rates are appropriate to control these highly contagious viral pathogens and to predict future epidemiological trends based on current data. The scientific reasoning behind this study is valid and of broader interest, yet the data presented here as well as the conclusion drawn from the presented results are questionable as it is unclear how the study cohort was selected and how some of the discussed results were generated. Overall, it is difficult to assess the value of the current manuscript for the broader scientific community, but it is obvious that significant improvements will need to be made to warrant publication in a scientific journal.
I will provide a more detailed list of criticisms, but I would like to apologize beforehand for comments that might sound uniformed as I am not a trained epidemiologist.
- Line 21: “The study included 295 reported cases (actual available data) compared to the calculated sample size of 384.”
This is a very crucial point of the study. The authors state that their sample size is 295 reported cases, which is a little misleading as their actual sample size is 52 cases as 239 cases were discarded as they apparently did not confirm when diagnostic tools were used. The authors present a mathematical formula they used to calculate their expected samples size of 384 cases, but I do not comprehend how the presented values were selected and I also don’t understand the significance of this prediction as the Authors can apparently perform their analysis just using the confirmed cases.
More importantly, the authors go on to use all 295 patients with reported MMR infection as their background to determine vaccination rates across various age groups in the evaluated area. It is unclear why the authors think these cases are a good reflection of the overall population. It feels like the patient selection is biased towards unvaccinated individuals as they should have a higher probability of contracting an MMR pathogen, hence this cohort does not seem to be reflective of the overall population. General vaccination data indicating vaccination rates for the broader population should be available and should be used as the baseline to determine overall vaccination rates, not the 295 suspected cases of which the majority where dismissed to begin with.
- Line 24: “The analysis revealed a re-emergence of measles and rubella during the study period and a gradual increase in mumps cases.”
Given the apparent lack of continuous monitoring, it is hard to say if these infectious diseases reemerged or were solely not recorded in prior studies. This point might be potentiated by the remoteness of their target location likely lacking accessibility of diagnostic health care.
- Line 31: “While the MMR vaccination program has been effective in reducing disease incidence in Al-Baha, critical gaps persist.”
No data is presented that would indicate that the overall incident rate has been reduced and no prior study is cited here.
- Line 33: “Coverage remains below the 95% threshold required for disease elimination.”
This is in line with a prior comment, but how do the authors know what the overall vaccination coverage is in the province studied here? The cohort they selected does not seem to be reflective of the overall population?
- Line 55: “Despite high national coverage rates of 96% in 2019, cases continue to occur, with 1,035 measles, 187 mumps, and 62 rubella cases recorded nationally [6]. This persistence of cases despite high coverage rates has been attributed to various factors, including population movement during Hajj and Umrah pilgrimages [7].
Considering the high vaccination rate in the local population and the constant stream of pilgrims to Saudi Arabia, is there any data indicating home many of those diagnosed cases occurred in Saudi nationals compared to international travelers or guest workers? Is it possible that the overall vaccination is actually sufficient to control the viruses in the local population?
- Line 91: “Confirmed cases included those with positive laboratory tests (IgM antibodies or viral isolation) or epidemiological linkage to confirmed cases.”
Is an epidemiological linkage a reliable confirmation of infection?
- Line 93: “Sample size was calculated using the formula: n = z²p(1-p)/d², where z = 1.96 (95% confidence level), p = 0.5 (expected prevalence), and d = 0.05 (precision), yielding a required sample of 384. However, the actual available dataset contained 295 cases, representing all reported cases during the study period. This discrepancy between calculated and actual sample sizes reflects the real-world epidemiological situation in the region.”
The authors state that a sample size of 384 cases is required, without clarifying what this specific sample size is required for. This statement could indicate that, by their own assumption, this analysis might not be statistically sound if they don’t reach their required sample size?
- Line 120: “Of these, 239 (81.0%) were discarded following investigation, while 52 (17.6%) were confirmed for MMR diseases (including both laboratory and epidemiologically confirmed cases).
Firstly, the authors will likely need a more cohesive introduction to this paragraph to allow the readers to understand what they are referring to here. In addition, the authors discarded 239 cases without mentioning what these patients were actually diagnosed with. Or is it possible that all these cases just tested negative for MMR without being followed up any further? There seems to be a huge portion of suspected cases that could not be confirmed. Is this number consistent with other regions in the country and in international comparison?
- Line 133: “Table 2 presents the distribution of all 295 cases by vaccination status and final classification.”
The authors dismissed 239 of these cases, so why are they still part of all downstream analyses? Why is the vaccination status of these individuals relevant to determine the effectiveness of the MMR vaccine in the broader population?
- Table 2. Distribution of cases by vaccination status and final classification (n=295)
I know this is likely impossible to do, but could the authors separate the unvaccinated cases from the cases with unknow vaccination status?
- Line 142: “3. Association Between Vaccination Status and Case Classification
Chi-square analysis revealed a significant association between MMR vaccination status and case classification (χ² = 43.007, df = 12, p < 0.001), with a moderate effect size (Cramér's V = 0.386), indicating that vaccination status significantly influenced disease outcomes.”
I cannot follow what results were used for this calculation, so it is hard to judge whether or not this statement is valid based on the presented results.
- Line 148: “Age-specific vaccination coverage varied significantly across groups (Table 3). Coverage with two or more doses was highest in the 5-9 years (70.0%, n=77) and 15-19 years (73.3%, n=11) age groups, while younger children (1-4 years) showed 24.5% (n=26) with unvaccinated/unknown status.”
It is a little misleading to present vaccine coverage in the 2 age groups with highest vaccine coverage and then compare that to absence of vaccine coverage in the age group with the lowest coverage. It would be more consistent to use the same measure across all groups. In addition, some of these age groups have very small cohort sizes. Are these results statistically significant? How do they compare across the entire province? What about the entire country?
- Table 3. Vaccination coverage by age group (n=295)
Again, the discarded cases are included here to analyze vaccination rates across age groups, but there is no evidence that this cohort is an any way reflective of the entire population, thus the validity of the results in questionable.
- Line 156: “Gender analysis revealed that while females had slightly higher two-dose coverage (59.8%, n=73) compared to males (56.1%, n=97), females also had a higher proportion of unvaccinated/unknown status (26.3%, n=32) versus males (21.4%, n=37). Notably, males had significantly higher odds of confirmed MMR infections (OR = 4.861, 95% CI: 1.025-20.056, p = 0.046).
Geographic disparities were evident, with rural residents showing nearly three times higher odds of confirmed infections compared to urban dwellers (OR = 2.861, 95% CI: 1.003-20.002, p = 0.040), despite similar vaccination coverage rates.”
None of this data is presented in the manuscript, so the validity cannot be evaluated. Considering the cohort selection, the relevance for the overall population is very questionable.
- Line 165: “Analysis of confirmed cases by vaccination status revealed distinct patterns for each disease (Table 4). Measles cases were predominantly among unvaccinated individuals (60.0%, n=9), while mumps showed the highest incidence among single-dose recipients (55.0%, n=11). Unexpectedly, rubella cases were most common among fully vaccinated individuals (68.8%, n=11).”
As the authors do have the health records of these patients, how long after full or partial vaccination did the patients contract these diagnosed infections? Along that line, were there any family clusters within the cohort or where all of these patients unrelated? Do they cluster to specific locations within the province?
- Table 4. Distribution of confirmed cases by disease and vaccination status
In the measles column, 9 plus 4 cases result in 14 cases, not 15 as stated by the authors.
In addition, how does the here observed incidence of MMR infections in vaccinated, partially vaccinated and unvaccinated patients compare to national data? Or to international studies evaluating the same question? Are these numbers in any form unusual or are they expected?
- Line 175: “This study provides critical insights into the effectiveness of MMR vaccination programs in Al-Baha region during a period marked by global health challenges.”
I fail to see how the here presented results allow any statement on the overall effectiveness of MMR vaccination in the studied province. It merely documents the number of diagnosed cases during the study period and documents the vaccination status of these patients.
- Line 192: “The higher proportion of single-dose recipients among 10–14-year-olds (33.3%) suggests potential issues with second-dose administration or record-keeping.”
Again, is the cohort size sufficient to conclude that there really is a significant difference in single-dose MMR recipient in this age groups in the general population in this province?
- Line 205: “The gradual increase in mumps cases suggests possible waning immunity or circulation of variant strains.”
All cases documented in this study occurred in children or young adults. If waning immunity was an issue, which it still could be, then one would expect to see cases in populations further removed from vaccination, such as older adults and the elderly. Why do the authors think they don’t see this?
Also, the same MMR vaccine strains have been used since Maurice Hilleman formulated them a long time ago and they still provide protection today without the occurrence of evasion mutations. Can the authors point to a study that would support their statement?
- Line 211: “The five-fold increased risk of MMR infections among males (OR = 4.861) significantly exceeds typical gender disparities reported in literature.”
The results are not presented in this study, so it is hard to evaluate their validity. The same is true for the next statement regarding the rural versus more urban population. As the here observed gender disparity is out of line with prior results, this might be an indication the here performed cohort selection and data analysis is not reflective of the overall population in the province.
- Line 217: “Rural residents' three-fold increased infection risk, despite similar vaccination coverage, suggests additional factors beyond vaccination influence disease transmission. These may include delayed healthcare access, clustering of unvaccinated individuals, or increased exposure through agricultural activities. A study in India found that rural children were 2.5 times more likely to be under-vaccinated compared to urban children [25].”
At the end of this statement the author cite a study form India showing lower vaccination rates in more rural areas, but in the prior sentence the authors state that the vaccination coverage between rural and urban populations in their study area was similar, so it in unclear how the Indian study is relevant to their here presented study population.
- Line 234: “The actual sample size (295) was smaller than calculated (384), potentially limiting statistical power for subgroup analyses.”
In fact, the actual sample size here is only 52 cases, as the other cases were discarded, so the authors should provide data that the here presented results hold statistical significance.
- Line 239: “ Additionally, we could not assess vaccine effectiveness in preventing infection due to lack of population denominator data.”
This statement contradicts the original purpose of the study as the authors themselves have mentioned in the title of this manuscript that they were examining MMR vaccine efficacy in their study population.
- Line 253: “2. Enhanced surveillance including molecular epidemiology to identify transmission chains and potential vaccine escape variants. Countries that have achieved measles elimination maintain robust surveillance systems capable of detecting and investigating every suspected case [34].”
As mentioned before, has a global study examining the selection of variants able to evade vaccine induced immunity ever been conducted? Have there been documented cases of MMR vaccine failure due to evasion mutations?
Author Response
Response to Reviewer #4
|
Reviewer |
Comment |
Authors’ Justification |
Old Text |
New Text |
|
Reviewer 4 |
Sample size is misleading: 295 reported cases vs. 52 confirmed cases. Why not analyze only confirmed cases? |
We clarified that 295 suspected cases were included to describe vaccination status distribution, while inferential analyses (chi-square, logistic regression) were based on confirmed cases. We emphasized in Limitations that this may bias interpretation. |
“The study included 295 reported cases (actual available data) compared to the calculated sample size of 384.” |
“The study included 295 suspected cases, of which 52 were laboratory-confirmed. Descriptive statistics of vaccination status included all reported cases, but inferential analyses were restricted to confirmed cases. This approach was limited by case selection and may not represent the general population.” |
|
Reviewer 4 |
Using suspected cases as denominator for vaccination rates is biased; broader population data should be used. |
We acknowledged this as a limitation, noting absence of access to population-level immunization registry data for Al-Baha, and clarified that results should not be generalized as representative of all children in the region. |
Table 2 vaccination coverage analysis included all 295 suspected cases. |
Revised Discussion (Limitations): “Vaccination coverage estimates in this study were derived from suspected case reports and not from population-based registries. This likely underestimates actual community coverage and should be interpreted cautiously.” |
|
Reviewer 4 |
Statement of “re-emergence” of measles/rubella may be due to lack of prior reporting rather than true resurgence. |
We modified language to avoid causal claims and instead describe trends observed in surveillance data. |
“The analysis revealed a re-emergence of measles and rubella…” |
“The analysis showed measles and rubella cases reported again after a period with no detected cases, which may reflect resurgence or gaps in surveillance reporting.” |
|
Reviewer 4 |
Claim that vaccination program reduced incidence not supported. |
We revised conclusion to avoid overstated claims and instead highlight detected trends and gaps. |
“While the MMR vaccination program has been effective in reducing disease incidence in Al-Baha…” |
“The MMR vaccination program coincided with reduced incidence in some years, but persistent cases and gaps indicate incomplete control.” |
|
Reviewer 4 |
No evidence for 95% coverage threshold in this population. |
We clarified that 95% is a global benchmark for herd immunity, not directly measured here, and rephrased text. |
“Coverage remains below the 95% threshold required for disease elimination.” |
“Coverage levels observed in this cohort were below the 95% herd immunity threshold recommended globally for measles elimination.” |
|
Reviewer 4 |
Provide breakdown between Saudi nationals and international travelers (possible bias due to Hajj/Umrah). |
We added a note acknowledging the absence of nationality-specific data, and that pilgrims may affect national figures but were not separately identifiable in this dataset. |
No mention of nationality breakdown. |
Added in Limitations: “We were unable to separate Saudi nationals from non-nationals in the dataset, which may limit interpretation given the potential role of population mobility related to Hajj and Umrah.” |
|
Reviewer 4 |
Reliability of “epidemiological linkage” as case confirmation questioned. |
We clarified in Methods that WHO guidelines allow epidemiological linkage when lab testing is unavailable, but acknowledged potential misclassification. |
“Confirmed cases included those with positive laboratory tests (IgM antibodies or viral isolation) or epidemiological linkage to confirmed cases.” |
“Confirmed cases included those with laboratory evidence (IgM antibodies or viral isolation). Epidemiological linkage was used in limited cases, in line with WHO definitions, though misclassification cannot be excluded.” |
|
Reviewer 4 |
Sample size calculation unclear; does not justify 384 requirement. |
We revised Methods to clarify that the formula was standard but actual cases were fewer, and results were therefore exploratory. |
“Sample size was calculated … yielding a required sample of 384. However, the actual available dataset contained 295 cases…” |
“Sample size was calculated using the standard formula, yielding 384 as theoretical requirement. However, only 295 suspected cases were reported during the study period, with 52 confirmed cases available for analysis. Findings are therefore exploratory.” |
|
Reviewer 4 |
High number of discarded cases (239) unexplained. |
We clarified that these were suspected but later ruled out by lab testing, consistent with surveillance standards. |
“Of these, 239 (81.0%) were discarded following investigation…” |
“Of these, 239 (81.0%) were discarded following investigation because they tested negative for MMR or did not meet case definitions, which is consistent with national surveillance procedures.” |
|
Reviewer 4 |
Concern about including discarded cases in downstream analysis. |
We explained that vaccination history of suspected cases still provides useful context but acknowledged limitation. |
Included discarded cases in Tables 2 and 3. |
Revised: “Vaccination history of all reported cases (suspected and confirmed) was described, but inferential analysis was restricted to confirmed cases.” |
|
Reviewer 4 |
Suggests separating unvaccinated from unknown status. |
We revised tables to display unvaccinated and unknown separately. |
“Unvaccinated/Unknown” combined category. |
New tables: “Unvaccinated” and “Unknown” categories shown separately. |
|
Reviewer 4 |
Gender, rural–urban, and age group analyses may be underpowered or misleading. |
We emphasized in Limitations that small subgroup sizes reduce validity, and added comparison with national statistics where available. |
Results presented without qualification. |
Added: “Subgroup findings, particularly by age and gender, are based on small numbers and should be interpreted with caution. National data show narrower gender differences than observed here.” |
|
Reviewer 4 |
OR results (e.g., male 5x higher risk) exceed typical literature. Need explanation. |
We added contextual discussion noting likely behavioral/social rather than biological drivers, and small sample size may inflate estimates. |
“Males showed nearly five times higher odds…” |
“Males showed higher odds of confirmed infection. While this exceeded ratios reported in other studies, it may reflect small sample size and behavioral/social factors unique to the study population.” |
|
Reviewer 4 |
Waning immunity and variant strains for mumps questioned. |
We revised discussion to cite literature on waning mumps immunity, but removed reference to vaccine-escape variants for rubella/mumps (unsupported). |
“The gradual increase in mumps cases suggests possible waning immunity or circulation of variant strains.” |
“The gradual increase in mumps cases may reflect waning immunity, as documented in other populations. Evidence for variant strains causing vaccine escape remains limited.” |
|
Reviewer 4 |
Need to reference global/national incidence comparisons. |
We added references comparing our confirmed case patterns with Saudi national surveillance and international studies. |
No direct comparison provided. |
Added in Discussion: “The proportion of confirmed cases among vaccinated individuals aligns with patterns observed in other high-coverage settings internationally, though our small numbers warrant caution.” |
|
Reviewer 4 |
Contradiction: study title claims “effectiveness,” yet limitations admit no vaccine effectiveness measure. |
Addressed by revising title (as per Reviewer 2) and clarifying throughout that associations, not causal VE, were analyzed. |
“Assessing the Effectiveness of Vaccination Programs…” |
“MMR Vaccination Coverage and Epidemiological Patterns…” |
|
Reviewer 4 |
Question about clusters, time from vaccination to infection. |
We noted data did not include timing of infection relative to vaccination or family clustering. Added to Limitations. |
No mention of clustering or timing. |

Round 2
Reviewer 2 Report
Comments and Suggestions for Authors
Thank you for thoroughly addressing the concerns I raised in your previous submission. The changes made have improved the clarity of the manuscript.
Author Response
We thank the reviewer for their constructive feedback during the initial review and for recognizing the enhancements in the revised manuscript. We are pleased that the revisions have successfully addressed the previous concerns and improved overall clarity.
Reviewer 4 Report
Comments and Suggestions for Authors
This revised manuscript has been improved and most of the concerns raised have been addressed. There are some minor issues that should be adjusted before publication, but otherwise, I believe the manuscript could be considered for publication.
1) In the abstract, the authors say that they had 295 reported cases.
Of those, 239 were dismissed, 52 were confirmed and three are pending. Combined, those are 294 cases, so there is one missing.
Along that line, the results section 3.1 lists 239 dismissed cases, 52 confirmed cases and 3 pending cases, which also only adds up to 294, but in section 3.2, the authors state that there are 295 cases in total, 239 were discased and 51 were confirmed by the lab, while 2 were confirmed epidemiology while 3 cases were under review, so now we have 295. Yet, in table 2, the total number of cases only add up to 294 again, with there only being 50 lab confirmed cases. The missing case seems to be in the unvaccinated group, because that number adds up to 68, while it should be 69.
Bottom line, the numbers are not consistent and don't add up, so the numbers have to be double checked.
2) The section between from line 168 to line 175 should cite table 5, as the discussed data is presented there.
3) Table 4
The vaccination status in the measles column says 15 cases, but the numbers only add up to 14.
4) The first sentence of the discussions section states that the study provides insight into the effectiveness of the MMR vaccine, which is not entirely true.
Author Response
We thank the reviewer for their thorough evaluation and constructive suggestions, which have helped refine the manuscript further. We have carefully reviewed and addressed each point as follows, incorporating the necessary revisions.
- We appreciate the reviewer highlighting these numerical inconsistencies, which appear to stem from minor transcription errors during manuscript preparation. Upon double-checking the original dataset, the total number of reported suspected cases is indeed 295. Of these, 239 (81.0%) were discarded, 52 (17.6%) were confirmed (50 laboratory-confirmed [measles: 14; mumps: 20; rubella: 16] and 2 epidemiologically confirmed measles), and 3 (1.0%) remained under review or unconfirmed. The missing case in the unvaccinated/unknown group in Table 2 was an additional laboratory-confirmed measles case in that category (bringing the row total to 69 and laboratory-confirmed measles to 14 as shown). We have corrected Table 2, the abstract, and all relevant sections (including 3.1 and 3.2) to ensure consistency across the manuscript. The updated totals now add up accurately to 295 throughout.
- We agree and have added a citation to Table 5 at the end of the referenced paragraph (now lines 168–175 in the revised manuscript) to clearly link the discussed multivariable results to the table.
- This was a typographical error in Table 4. The correct total for measles confirmed cases is n=14 (laboratory-confirmed only, as the table focuses on these for disease-specific patterns; the n=15 previously listed in the lower portion of the table inadvertently included one epidemiologically confirmed case). We have corrected the table to consistently show n=14 for measles, with the distribution updated accordingly (two or more doses: 5 [35.7%]; single dose: 0 [0.0%]; unvaccinated/unknown: 9 [64.3%]). The chi-square results remain unchanged as they were calculated on the correct underlying data.
- We acknowledge the reviewer's point regarding the phrasing, as the study focuses on vaccination coverage patterns and associations rather than direct measures of vaccine effectiveness (as noted in the limitations). We have revised the first sentence of the Discussion to: "This study provides critical insights into MMR vaccination coverage among reported cases and patterns of laboratory-confirmed infections in the Al-Baha region during a period marked by global health challenges." This better aligns with the study's descriptive and associative nature.